# Reply to Kaminski, N.E.; Cohen, S.M. Comment on “Bischoff et al. The Effects of the Food Additive Titanium Dioxide (E171) on Tumor Formation and Gene Expression in the Colon of a Transgenic Mouse Model for Colorectal Cancer. *Nanomaterials* 2022, *12*, 1256”

**DOI:** 10.3390/nano13091552

**Published:** 2023-05-05

**Authors:** Nicolaj S. Bischoff, Héloïse Proquin, Marlon J. Jetten, Yannick Schrooders, Marloes C. M. Jonkhout, Jacco J. Briedé, Simone G. van Breda, Danyel G. J. Jennen, Estefany I. Medina-Reyes, Norma L. Delgado-Buenrostro, Yolanda I. Chirino, Henk van Loveren, Theo M. de Kok

**Affiliations:** 1Department of Toxicogenomics, GROW School for Oncology and Reproduction, Maastricht University Medical Center, 6229 ER Maastricht, The Netherlands; 2National Institute for Public Health and Environment (RIVM), 3721 MA Bilthoven, The Netherlands; 3Faculty of Health, Medicine and Life Science, Maastricht University Medical Center, 6229 ES Maastricht, The Netherlands; 4Laboratory of Biosignaling & Therapeutics, Department of Cellular and Molecular Medicine, KU Leuven, 3000 Leuven, Belgium; 5Laboratorio de Carcinogénesis y Toxicología, Unidad de Biomedicina, Facultad de Estudios Superiores Iztacala, Universidad Nacional Autónoma de México, Mexico City 54090, Mexico

We appreciate the interest in our article describing transcriptome changes in a transgenic mouse model carrying an APC gene mutation and would like to reply to the reader [1]. Our paper focuses primarily on the transcriptome analysis while providing additional information on tumor formation and histological changes. The results presented in this publication clearly show a significantly altered gene expression following exposure to food-grade E171. We would like to emphasize that the experiment on the effects of E171 on tumor formation was simply done to examine if the exposure to E171 in this transgenic mouse enhances tumor growth/formation and compare these results to studies in wild-type animals, as published by our group before [2,3]. There are differences, notably in the magnitude of the stimulation, which was much more modest (Table 2 main manuscript). We did not indicate at any point that the formation of tumors as such was statistically significant, while tumor size showed a statistically significant increase. Regarding the transcriptome data; these were checked for quality and normalized with the ArrayQC pipeline (GitHub—arrayanalysis/arrayQC_ModuleGitHub—arrayanalysis/arrayQC_Module) as previously published by our department. All results from the transcriptome experiments that were statistically significant have been checked for FDR and displayed accordingly, as clearly stated throughout the paper. All of the statistical methods and cut-off values (FC < −1.5, FC > 1.5, *p*-value < 0.05, and q-value > 0.05) for the transcriptome analysis are described in great detail in the Materials and Method section of the main manuscript (Sections 2.8, 2.9, 2.10 and 2.11).

Regarding the route of administration and composition of the administrated titanium dioxide, we would like to point out that our experiments were carried out via drinking water in the pilot study and later on via an oral gavage to account for potential discrepancies in daily dosage when given via drinking water. While this does not mimic the human situation exactly, it allows for more accurate dosing and controlled exposure. The food-grade titanium dioxide which was used in our study was obtained from MARK al Chemical, Mexico, and represents a product that is commercially available on the market and labelled food-grade. In regard to the range of nanoparticles, the publication by Weir et al., 2012, which is referenced in the text details that there are at least 36% nanoparticles within the pristine materials [4]. More recent publications by Verleysen et al., 2020, also show that the percentage of nanoparticles in commercially available pristine E171 might be higher than previously estimated, ranging from 20 to 100% [5]. Our characterization via TEM and sp-ICP-MS showed a high percentage of nanoparticles within the pristine materials of up to ~64%. The Z-average of said preparations was additionally analyzed via DLS and showed an average size ranging from 315 to 350 nm of the titanium dioxide aggregates, with a zeta potential of −29.9 to −27.1 (pH ~7.3). These results indicate a rather stable dispersion, with the majority of particles being aggregated in larger clusters, which is in line with above-mentioned publication and displays the state of the particles upon administration via oral gavage.

We fully agree with the reader that the most accurate way of estimating hazard and risk for humans following the ingestion of food-grade titanium dioxide would be oral administration of E171 in a food matrix, preferably in humans. We are currently executing exactly such a human exposure study in our group.

We acknowledge the concerns raised regarding the use of this transgenic mouse model when examining tumor formation. We chose this specific model (transgenic mouse model carrying a heterozygous mutation of the APC gene) to be in line with a commonly found mutation in the early stage of colorectal cancer development. This allowed us to examine the effects of food grade E171 on the formation of tumors and gene expression changes in the colon of the animals. While our study did not show an increase in the number of tumors per animal, we demonstrated that the size of the tumors found in the group exposed to E171 was significantly increased in comparison with the controls. Furthermore, we demonstrated that exposure to food grade E171 leads to transcriptome changes. The latter were studied by examining the differentially expressed genes with various bioinformatics methods which described early transcriptional changes.

Following the letter to the editor, we examined our histopathology samples again with two independent histopathologists. Indeed, what we indicated as lymph nodules are not those structures, and we apologize for the misconception. However, neither are these lymphoid aggregates, as suggested by the reader. What we indicated as hyperplastic lymph nodules in Figure 2 (red squares, main manuscript) are areas of enhanced epithelial cells infiltrated in the muscle layer, showing an invasive process from epithelial cells, which also supports the fact that this tissue is an adenocarcinoma. In addition, we also observed nuclear enlargement and an increased nucleus-to-cytoplasmic ratio (anaplasia), which is characteristic of adenocarcinomas [6]. Based on these elements, the consulted histopathologist classified these polyps as adenocarcinomas. Below, we provide a new figure (Figure 1) including more detail. We will modify Section 3.2 and the consequent section in the discussion of the main manuscript and will submit the correction accordingly and promptly.

Furthermore, we would like to address the reader’s comment about the sessile serrated polyps. A polyp is an abnormal growth of tissue that protrudes from a mucous membrane. This process, by definition, implies hyperplasia. What we observed was large hyperplastic areas, showing increased cell proliferation but also infiltration of epithelial cells in the muscle layer, which denotes malignancy. In addition, we also observed a loss of tissue architecture, a nuclear enlargement, and an increased nucleus-to-cytoplasmic ratio (anaplasia) in E171-exposed mice, which is characteristic of adenocarcinomas [6]. Based on these elements, the histopathologist classified these polyps as adenocarcinomas. We would also like to mention that sessile serrated lesions are more common in the right colon [7,8] and the piece of colon analyzed in this work was the left (distal).

We strongly disagree with the statement that our Materials and Methods section, regarding the transcriptome study, is somehow misleading. The colonic material used for the transcriptome analysis was taken 2, 7, 14, and 21 days after exposure to E171. These time points were chosen to provide insights in the genetic response of E171 following an acute and subacute exposure. The insights obtained from such an early stage analysis were of interest to us to better understand the potential hazards arising from E171 exposure. Additionally, the preparation of the colon was described in great detail in the Material and Methods, particularly in Section 2.6: mRNA Extraction from Colonic Tissues: “As tumor formation in this mouse model was mainly found in the distal colon [16], mRNA was extracted from this part of the colon as previously reported”. The distal colon was used for microarray analyses and, according to the NIH definition of the distal colon, “the distal colon includes the descending colon (the left side of the colon) and the sigmoid colon (the S-shaped section of the colon that connects to the rectum)”. Therefore, only material originating from the left part of the colon was used for gene expression analyses. As all tumors detected were present in the distal colon, a section of this part of the colon can be considered as most appropriate. The colon was checked and tumors were not included in the analysis, neither were lymph nodes. The design of the study was meant to give a full overview of the gene expression after exposure to E171. This is based on a complex cell population (the submucosa and even in the lamina propria, with a large number of lymphocytes, macrophages, blood vessels, and fibroblasts, as well as the smooth muscle cells of the muscle wall) in order to have a comparable cell distribution pre- and post-exposure. Development of cancer is not only due to the epithelial cells but also to the environment, for example, the evasion of the immune system [9] by tumors is a known strategy that must be taken into account while studying cancer development, especially at an early stage. Therefore, to study how tumor formation can also escape the immune system, studying only epithelial cells is not sufficient. Hence, the target tissue in our study was not epithelium tissue, as this tissue is only a minor cellular component of the colon. The study was designed to integrate the gene expression response profile based on a complex cell population.

Furthermore, we would like to address the concerns regarding the accuracy of using microarrays for transcriptome analyses. Microarrays have been the gold standard in transcriptome analyses for years by being a solid and reliable tool, as well as being cost efficient and easy to use. They provide insights into hundreds of genetic and molecular changes and advanced the field of genomics/toxicogenomics immensely. Every technique has its up and downsides and is used for specific purposes. To account for potential false positive or false negative results, multiple correction steps are incorporated into the workflow, e.g., false discovery rate (FDR), data normalization, removal of bad spots with low expression, etc. Genes were analyzed by using the widely accepted workflow of LIMMA, which represents a linear model to assess differential expression in the context of multi-factor designed experiments, with cut-offs set at absolute FCs of >1.5 and <−1.5, in combination with a *p*-value of <0.05. Additionally, the results were corrected for FDR with an additional <0.05 threshold (q-value), as is the common practice. Pathway analyses via CPDB and STEM and network analysis via Metascape followed the same or even stricter correction thresholds, as indicated throughout the manuscript. The heatmap provided in Figure 7 displays all significantly altered genes at the different time points after ORA analysis in CPDB (LIMMA analysis with absolute FC = 1.5, q-value < 0.05). Table 3 summarizes the pathway analysis of these genes and does not mention any changes in pathways related to inflammation or cytokines. However, these particular pathways were identified thanks to the STEM analysis, which is based on different principles and is summarized in Figure 8 and Table 4. Furthermore, the genes that are differentially expressed in the respective clusters can be found in the tables in the supplementary materials. STEM is a tool for clustering and visualization of short time series microarray experiments. It identifies significant temporal expression profiles and the genes that are associated with them. These clustered genes were used for a pathway analysis (CPDB), and led to the findings including cytokine interactions, B cell signaling, and other immune regulatory processes. We are familiar with the work of your group and your above-mentioned publications. Your work focuses on endpoints, while this set of experiments mainly aims to identify early upstream events (hazard identification). Furthermore, the experimental design and execution are fundamentally different, and hence not directly comparable. While we did not examine any additional downstream endpoints such as interleukin levels in the bloodstream, your study did not examine gene expression in your animals; therefore, no comparison can be made between the two since they are addressing and focusing on different endpoints. Combining our transcriptome data with additional analyses of downstream endpoints would be the optimal scenario and vice versa.

We stated that no clear dose–response was observed over time when looking at the numbers of DEGs. This might be on the one hand related to a very narrow margin of doses that the animals were exposed to, and which mimic relevant human exposure levels (EFSA: maximum level exposure assessment scenario, adults mean: 1.3–6.2 mg/kg bw/day). On the other hand, the number of DEGs does give an overview of how many genes are differentially expressed but does not indicate the magnitude of the effect that is induced on those genes or the magnitude of differential expression. To keep the complex transcriptome data more comprehensive, we decided to focus on only one dose, which was chosen to be 1 mg/kg bw/day.

The Discussion in our paper is based on our observations, mainly from the transcriptome analysis, and describes early expression changes in the analyzed colon tissue and their potential connection to the development of colorectal cancer (CRC). The results of the tumor formation study are clearly described in the Results and Discussion sections and indicated that the increase in tumor number was not statistically significant. At the same time, all transcriptome results are based on a profound statistical analysis and correction with strict and generally accepted thresholds. These findings indicate early time points that can be linked to CRC development. No tumors are expected to form within the timeframe of the exposure in the transcriptome study, and no tumor tissue has been used for transcriptome analyses. We discussed the findings of our gene expression study and compared those findings with the literature on the relationship between the significantly altered gene expression of those genes and colorectal cancer. The network and interaction of the genes were analyzed and presented in the functional enrichment analysis, which confirmed the findings of the ORA and STEM analyses and highlighted the interconnection of the genes that were identified after the LIMMA analysis in Figure 10. The network shows genes that were differentially expressed and their network connection to each other, indicated by colors and positioning. None of the displayed results are solely based on a 1.5 FC of any gene but are always a combination of FC, *p*-values, FDR correction, STEM, or the gene enrichment analysis. While these transcriptome changes do not necessarily result in phenotypical changes, they are indicators of underlying molecular processes which were triggered by the exposure to E171. Therefore, they are providing valuable information to better understand potential hazards or modes of action.

We hope our insights made it easier to understand the complex matter of transcriptome analyses and helped to highlight the focus of our study. Certainly, this study did provide evidence for transcriptome and histological changes in the colon of this Tg mouse model. Throughout our publication, we show statistically significant findings on gene expression alteration and differences in tumor size. These findings and their potential association with early stage colorectal cancer development are described in detail and with references in the discussion section of this publication. Interpretation of transcriptome data is only one step in the process of hazard identification and additional follow-up experiments and validation of potential downstream endpoints certainly add value to the conducted study. This was unfortunately not possible in the framework of the presented workflow. Nevertheless, we are confident that the data presented in this publication are scientifically sound and a valuable contribution to the ongoing debate on the safety of E171 in food and feed.

## Figures and Tables

**Figure 1 nanomaterials-13-01552-f001:**
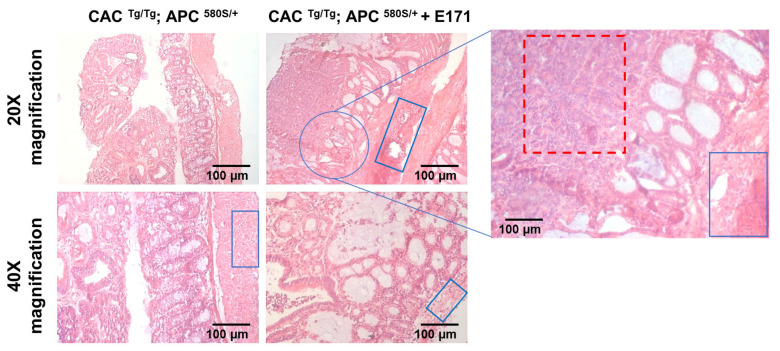
Representative images of colon histology. The hyperplastic area showing a loss of architecture and noticeable nucleus enlargement is indicated with a red dotted box. Infiltrated neoplastic epithelial cells in the muscle layer are indicated with blue boxes.

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
