# Peer review of "Reply to Kaminski, N.E.; Cohen, S.M. Comment on “Bischoff et al. The Effects of the Food Additive Titanium Dioxide (E171) on Tumor Formation and Gene Expression in the Colon of a Transgenic Mouse Model for Colorectal Cancer. Nanomaterials 2022, 12, 1256”"

_nanomaterials, 2023, doi:10.3390/nano13091552_

Round 1
Reviewer 1 Report
The Authors have properly replied to comments on their own manuscript.
Author Response
Dear esteemed Reviewer,
Thank you a lot for taking the time to revise our reply. We spelled-checked the manuscript once more and corrected the remaining misspellings.
Reviewer 2 Report
The Reply to Comment on the Paper “The Effects of the Food Additive Titanium Dioxide (C171) On Tumor Formation and Gene Expression in the Colon of a Transgenic Mouse Model for Colorectal Cancer Nanomaterials, 12(8):1256, 2022”m is timely, globally comprehensive and concise.
Suggestions are described below:
1) Was the data validated by statistical analysis? If so, therefore there is no need to refer that the values were “significantly”. However, the quantitative information about the decrease (or increase) is fundamental. The data should be described first. Therefore please avoid imprecise terms such as: significantly (But how much? 10%? 2-fold?); more modest (How modest?); statistically significant increase (But how much? From x to y would be more quantitative and precise); E171 was significantly increase (But how much? 200%?.
2) Fig S2: Scale bars should be inserting in all panels not only in one.
Author Response
Dear esteemed reviewer,
We thank you for taking the time to revise the manuscript and provide us with your feedback. We added the scale to all images, as suggested.
All statistical methods are described in the main manuscript. Tumor formation and tumor size were analyzed by one-way ANOVA with a set threshold for the p-values of 0.05. Furthermore, all microarray experiments that were examined, met the statistical cut-offs. These vary slightly between analyses but generally consist of FC of > +/- 1.5, p-value 0.05, q-value 0.05 (genes)/ 0.01 (pathways).
The "modest" response referenced in the introduction to our reply, refers to an overall comparison of this study with Tg mice, to previous experiments in our group in BALB/c mice and a DSS/AOM mouse model, which showed an overall larger response in altered pathways. The comparison to our previous studies was not statistically examined. A reference to the exact values as stated in Table 2 in the main manuscript has been added.